# TOWARDS NATURAL ROBUSTNESS AGAINST ADVERSARIAL EXAMPLES

## ABSTRACT

Recent studies have shown that deep neural networks are vulnerable to adversarial examples, but most of the methods proposed to defense adversarial examples cannot solve this problem fundamentally. In this paper, we theoretically prove that there is an upper bound for neural networks with identity mappings to constrain the error caused by adversarial noises. However, in actual computations, this kind of neural network no longer holds any upper bound and is therefore susceptible to adversarial examples. Following similar procedures, we explain why adversarial examples can fool other deep neural networks with skip connections. Furthermore, we demonstrate that a new family of deep neural networks called Neural ODEs (Chen et al., 2018) holds a weaker upper bound. This weaker upper bound prevents the amount of change in the result from being too large. Thus, Neural ODEs have natural robustness against adversarial examples. We evaluate the performance of Neural ODEs compared with ResNet under three white-box adversarial attacks (FGSM, PGD, DI$^2$-FGSM) and one black-box adversarial attack (Boundary Attack). Finally, we show that the natural robustness of Neural ODEs is even better than the robustness of neural networks that are trained with adversarial training methods, such as TRADES and YOPO.

## 1 INTRODUCTION

Deep neural networks have made great progress in numerous domains of machine learning, especially in computer vision. But Szegedy et al. (2013) found that most of the existing state-of-the-art neural networks are easily fooled by adversarial examples that generated by putting only very small perturbations to the input images. Since realizing the unstability of deep neural networks, researchers have proposed different kinds of methods to defense adversarial examples, such as adversarial training (Goodfellow et al., 2014), data compression (Dziugaite et al., 2016), and distillation defense (Papernot et al., 2016). But each of these methods is a remedy for the original problem, and none of these methods can solve it fundamentally. For example, Moosavi-Dezfooli et al. (2016) showed that no matter how much adversarial examples are added to training sets, there are new adversarial examples that can successfully attack the adversarial trained deep neural network.

So, avoiding adversarial examples technically cannot solve the most essential problem: why such subtle change in adversarial examples can beat deep neural networks? Meanwhile, it leads to a more important question: how to make deep neural networks have natural robustness so that they can get rid of malicious adversarial examples.

Early explanations for adversarial examples considered that a smoothness prior is typically valid for kernel methods that imperceptibly tiny perturbations of a given image do not normally change the underlying class, while the smoothness assumption does not hold for deep neural networks due to its high non-linearity (Szegedy et al., 2013). This analysis underlies plain deep neural networks like AlexNet (Krizhevsky et al., 2012). But later than that, Goodfellow et al. (2014) claim adversarial examples are a result of models being too linear rather than too non-linear, they can be explained as a property of high-dimensional dot products. Unfortunately, both of these explanations seem to imply that adversarial examples are inevitable for deep neural networks.

On the other hand, we notice that skip connections are widely used in current deep neural networks after the appearance of Highway Network (Srivastava et al., 2015) and ResNet (He et al., 2016). It turns out that the identity mapping in ResNet is formally equivalent to one step of Euler's method

which has been used to solve ordinary differential equations (Weinan, 2017). More than that, other kinds of skip connections used by different network architectures can be considered as different numerical methods for solving ordinary differential equations. The link between numerical ordinary differential equations with deep neural networks can bring us a whole new perspective to explain adversarial examples through the numerical stability analysis.

In this paper, we attempt to utilize the natural property of neural networks to defense adversarial examples. We first analyze how adversarial examples affect the output of neural networks with identity mappings, obtain an upper bound for this kind of neural networks, and find that this upper bound is impractical in actual computations. In the same way, we figure out why adversarial examples can fool commonly used deep neural networks with skip connections. Then, we demonstrate that Neural ODEs hold a weaker upper bound and verify the natural robustness of Neural ODEs under four types of perturbations. Finally, we compare Neural ODEs with three types of adversarial training methods to show that the natural robustness of Neural ODEs is better than the robustness of neural networks that are trained with adversarial training. The main contributions of our work are as follows:

- We introduce and formalize the numerical stability analysis for deep neural networks with identity mappings, prove that there is an upper bound for neural networks with identity mappings to constrain the error caused by adversarial noises.

- We provide a new reason why commonly used deep neural networks with skip connections cannot resist adversarial examples.

- We demonstrate that Neural ODEs hold a weaker upper bound which limits the amount of change in the result from being too large. Compare with ResNet and three types of adversarial training methods, we show the natural robustness of Neural ODEs.

## 2 RELATED WORKS

### 2.1 ADVERSARIAL DEFENSE

Adversarial training typically uses a robust optimization to generate adversarial examples for training deep neural networks. Madry et al. (2017) take the optimization as a saddle point problem. Zhang et al. (2019a) cast adversarial training as a discrete time differential game. Adversarial training can be seen as a data augmentation particularly enhance the robustness to white-box attacks (Tramèr et al., 2017). Zantedeschi et al. (2017) augmented the training sets with examples perturbed using Gaussian noises which can also enhance the robustness to black-box attacks. Lee et al. (2017) proposed a novel adversarial training method using a generative adversarial network framework. Besides, Finlay et al. (2018) augmented adversarial training with worst case adversarial training which improves adversarial robustness in the $\ell_2$ norm on CIFAR10.

Modifying the neural networks by using auto encoders, input gradient regularization and distillation can result in robustness against adversarial attacks (Bai et al., 2017; Ross & Doshi-Velez, 2017; Papernot et al., 2016). Besides, there are some biologically inspired deep learning models designed to have natural robustness against adversarial examples. Nayebi & Ganguli (2017) developed a scheme similar to nonlinear dendritic computation to train deep neural networks to make them robust to adversarial attacks. Krotov & Hopfield (2018) proposed Dense Associative Memory (DAM) models and suggested that DAM with higher order energy functions are closer to human visual perception than deep neural networks with ReLUs.

In addition to augmenting datasets or modifying original neural networks, there exist adversarial defense methods that rely on using external models and detecting adversarial examples. Akhtar et al. (2018) presented Perturbation Rectifying Network (PRN) as 'pre-input' layers to a targeted model, if a perturbation is detected, the output of the PRN is used for label prediction instead of the actual image. Xu et al. (2017) proposed a strategy called feature squeezing to reduce the search space available to an adversary by coalescing samples that correspond to many different feature vectors in the original space into a single sample.

## 3 NUMERICAL STABILITY ANALYSIS FOR DNNS

### 3.1 RESNET AND EULER'S METHOD

The building block of ResNet is defined as

$$\mathbf{y}_{n+1} = \mathbf{y}_n + f(\mathbf{y}_n; \theta_n) \tag{1}$$

Where $n \in \{0...N(h) - 1\}$, $\mathbf{y}_n$ and $\mathbf{y}_{n+1}$ are input and output vectors of the layers considered. The function $f(\mathbf{y}_n; \theta_n)$ represents the residual mapping and $\theta_n$ are weights to be learned.

In numerical methods for solving ordinary differential equations, Euler's method is defined by taking this to be exact:

$$\mathbf{y}_{n+1} = \mathbf{y}_n + hf(t_n, \mathbf{y}_n; \theta_n) \tag{2}$$

It can be easily seen that Eqn.(1) is a special case of Euler's method when the step size $h = 1$. The iterative updates of ResNet can be seen as an Euler discretization of a continuous transformation (Lu et al., 2018; Haber & Ruthotto, 2017; Ruthotto & Haber, 2019). When the input data is perturbed by adversarial noises $\epsilon$, we denote the adversarial example by

$$\mathbf{z}_0 = \mathbf{y}_0 + \epsilon \tag{3}$$

To perform stability analysis for ResNet with adversarial example $\mathbf{z}_0$ as input, we define a numerical solution $\mathbf{z}_n$ by

$$\mathbf{z}_{n+1} = \mathbf{z}_n + hf(t_n, \mathbf{z}_n; \theta_n) \tag{4}$$

and provide following theorem to show that the amount of change between $\mathbf{y}_n$ and $\mathbf{z}_n$ holds an upper bound by some assumptions.

**Theorem 3.1.** *Let $f(t, \mathbf{y}; \theta)$ be a continuous function for $t_0 \leq t \leq b$ and $-\infty < \mathbf{y} < \infty$, and further assume that $f(t, \mathbf{y}; \theta)$ satisfies the Lipschitz condition. Then compare the two numerical solutions $\mathbf{y}_n$ and $\mathbf{z}_n$ as $h \to 0$, there is a constant $c \geq 0$, such that amount of change between $\mathbf{y}_n$ and $\mathbf{z}_n$ satisfies*

$$\max_{0 \leq n \leq N(h)} |\mathbf{z}_n - \mathbf{y}_n| \leq c|\epsilon| \tag{5}$$

*Proof.* Let $\mathbf{e}_n = \mathbf{z}_n - \mathbf{y}_n, n \geq 0$. Then $\mathbf{e}_0 = \epsilon$, and subtracting Eqn.(2) from Eqn.(4), we obtain

$$\mathbf{e}_{n+1} = \mathbf{e}_n + h[f(t_n, \mathbf{z}_n; \theta_n) - f(t_n, \mathbf{y}_n; \theta_n)] \tag{6}$$

Assume that the derivative function $f(t, \mathbf{y}; \theta)$ satisfied the following Lipschitz condition: there exists $K \geq 0$ such that

$$|f(t, \mathbf{y}_1; \theta) - f(t, \mathbf{y}_2; \theta)| \leq K|\mathbf{y}_1 - \mathbf{y}_2| \tag{7}$$

Taking bounds using Eqn.(7), we obtain

$$|\mathbf{e}_{n+1}| \leq |\mathbf{e}_n| + h\hat{K}|\mathbf{z}_n - \mathbf{y}_n| \tag{8}$$

$$|\mathbf{e}_{n+1}| \leq (1 + h\hat{K})|\mathbf{e}_n| \tag{9}$$

where $\hat{K}$ is the largest Lipschitz constant.

Apply this recursively to obtain

$$|\mathbf{e}_n| \leq (1 + h\hat{K})^n |\mathbf{e}_0| \tag{10}$$

Using the inequality $(1 + t)^m \leq e^{mt}$, for any $t \geq -1$, any $m \geq 0$, we obtain

$$(1 + hK)^n \leq \mathbf{e}^{nh\hat{K}} = \mathbf{e}^{(b-t_0)\hat{K}} \tag{11}$$

and this implies the main result Eqn.(5). □

Roughly speaking, theorem 3.1 means that a small adversarial perturbation initial value of the problem leads to a small change in the solution, provided that the function $f(t, \mathbf{y}; \theta)$ is continuous and the step size $h$ is sufficiently small.

Obviously, ResNet does not satisfied the assumption of continuity and its step size $h$ always equal to 1. Particularly, the functions $f(t, \mathbf{y}; \theta)$ in ResNet are composite function which contains ReLU

activation functions. Although ReLU activation functions $g(x) = max(0, x)$ break the continuity, due to their contractive property, i.e. satisfies $\|g(x) - g(x + \epsilon)\| \leq |\epsilon|$ for all $x, \epsilon$; it follows that

$$
\begin{aligned}
&\|g_n(x; \theta_n) - g_n(x + \epsilon; \theta_n)\| \\
&= \|max(0, \theta_n x + b_n) - max(0, \theta_n(x + \epsilon) + b_n))\| \\
&\leq \|\theta_n \epsilon\| \leq \|\theta_n\| \|\epsilon\|
\end{aligned}
\tag{12}
$$

This provides a weaker upper bound for ReLU to mitigate the loss of continuity.

Even if the step size $h$ in ResNet can be adjusted by multiply outputs of convolution layers by a chosen constant, but unfortunately, the step size $h$ in ResNet cannot be too small since a very small step size decreases the efficiency in actual computations. We can experimentally show that when step size $h$ is small (such as $10^{-1}$, $10^{-2}$ and $10^{-3}$), ResNet has no obvious robustness against adversarial examples, and when step size is very small (such as $10^{-8}$, $10^{-9}$, $10^{-10}$), ResNet is difficult to train, so not only the classification accuracy of adversarial examples but also the accuracy of clean examples is quite low.

To summarize, the main reason for ResNet's failure in adversarial examples is the step size $h = 1$ destroys the upper bound given by Eqn.(5) and we cannot find a proper way to deal with it. Thus, when the input of ResNet is perturbed by adversarial noises, the amount of change in the result will become unpredictably large so that ResNet can no longer correctly classify the input.

### 3.2 Neural ODEs

The residual block can be described as $\mathbf{y}_{n+1} = \mathbf{y}_n + hf(\mathbf{y}_n; \theta_n)$ with step size $h = 1$. When taking smaller steps and add more layers, which in the limit, Neural ODEs parameterize the continuous dynamics of hidden units using an ordinary differential equation specified by a neural network

$$
\lim_{h \to 0} \frac{\mathbf{y}_{n+h} - \mathbf{y}_n}{h} = \frac{d\mathbf{y}}{dh} = f(t, \mathbf{y}; \theta)
\tag{13}
$$

The solution of Eqn.(13) can be computed using modern ODE solvers such as Runge–Kutta (Runge, 1895; Kutta, 1901). Euler's method only uses $f(t_n, \mathbf{y}_n; \theta_n)$, which means from any point on a curve, we can find an approximation of a nearby point on the curve by moving a short distance along a tangent line to the curve. Instead of using a single tangent line, modern numerical ODE methods often use lots of different tangent lines and weighted sum all of them to generate a super tangent line, we denote the super tangent line by $F(t_n, \mathbf{y}_n; \theta_n, f)$.

We can show that modern ODE solvers based on Runge–Kutta are stable under some assumptions, the amount of change between $\mathbf{y}_n$ and $\mathbf{z}_n$ holds an upper bound similar to theorem 3.1.

**Theorem 3.2.** *Let $f(t, \mathbf{y}; \theta)$ be a continuous function for $t_0 \leq t \leq b$ and $-\infty < \mathbf{y} < \infty$, and further assume that $f(t, \mathbf{y}; \theta)$, $F(t, \mathbf{y}; \theta, f)$ satisfies the Lipschitz condition. Then compare the two numerical solutions $\mathbf{y}_n$ and $\mathbf{z}_n$ as $h \to 0$, there is a constant $\hat{c} \geq 0$, such that amount of change between $\mathbf{y}_n$ and $\mathbf{z}_n$ satisfies*

$$
\max_{0 \leq n \leq N(h)} |\mathbf{z}_n - \mathbf{y}_n| \leq \hat{c} |\epsilon|
\tag{14}
$$

*Proof.* This result can be obtained in analogy with Eqn.(5) in theorem 3.1. $\qquad\square$

The neural network in Neural ODEs also use ReLU activation functions. And we already show that Eqn.(12) provides a weaker upper bound for ReLU.

Besides, modern ODE solvers, such as Fehlberg method (Fehlberg, 1969) and DOPRI method (Dormand & Prince, 1980), can adaptively adjust the step size until that the desired tolerance is reached. At each step, two different approximations (for example, a fourth order Runge–Kutta and a fifth order Runge–Kutta) for the solution are made and compared. If the two answers are in close agreement, the approximation is accepted. If the two answers do not agree to a specified accuracy, the step size is reduced. If the answers agree to more significant digits than required, the step size is increased. So modern ODE solvers provide a weaker upper bound for step size when it does not agree with $h \to 0$.

To summarize, although Neural ODEs cannot hold the strong upper bound provided by Eqn.(14), it still holds a weaker upper bound to constrain the error caused by adversarial noises. We will experimentally show that Neural ODEs is more robust on adversarial examples compared with ResNet.

### 3.3 OTHER DEEP NEURAL NETWORKS WITH SKIP CONNECTIONS

Following similar procedures, we can figure out the reason why other deep neural networks with skip connections cannot resist adversarial examples.

#### 3.3.1 POLYNET

PolyNet (Zhang et al., 2017) introduced a family of models called PolyInception. Each PolyInception module is a polynomial combination of Inception units that can be described as

$$\left(I + F + F^2\right) \cdot \mathbf{y} = \mathbf{y} + F(\mathbf{y}) + F(F(\mathbf{y})) \tag{15}$$

Where $\mathbf{y}$ denotes the input, $I$ the identity operator, and $F$ the nonlinear transform carried out by the residual block, which can also be considered as an operator.

It has been shown that Eqn.(15) can be interpreted as an approximation to one step of the backward (or implicit) Euler method (Lu et al., 2018):

$$\mathbf{y}_{n+1} = \mathbf{y}_n + hf(t_{n+1}, \mathbf{y}_{n+1}) \tag{16}$$

And from Eqn.(16) we can get

$$\mathbf{y}_{n+1} = (I - hf)^{-1}\mathbf{y}_n \tag{17}$$

Where $(I - hf)^{-1}$ can be formally rewritten as

$$I + hf + (hf)^2 + \cdots + (hf)^n + \cdots \tag{18}$$

So, there are two differences between the polynomial combination with the backward Euler method:

- The polynomial combination is only a second order approximation to the backward Euler method, the truncation error generated when high order terms are ignored.
- PolyNet faces the same problem with ResNet that the step size $h = 1$. Thus, PolyNet cannot holds the upper bound given by the backward Euler method.

And step size $h = 1$ is the main reason that PolyNet fails to resist adversarial examples.

#### 3.3.2 FRACTALNET

FractalNet (Larsson et al., 2016) introduced a design strategy for neural network macro-architecture based on self-similarity. The expansion rule generates a fractal architecture with $C$ intertwined columns can be described as

$$f_{C+1}(\mathbf{y}) = \frac{1}{2}(f_C \circ f_C)(\mathbf{y}) + \frac{1}{2}f_1(\mathbf{y}) \tag{19}$$

Where $f_1(\mathbf{y}) = \text{conv}(\mathbf{y})$ and $\circ$ denotes composition.

This expansion rule resembles to the Runge-Kutta method of order 2 (also known as Heun's method)(Lu et al., 2018)

$$\mathbf{y}_{n+1} = \mathbf{y}_n + \frac{1}{2}h(f(t_n, \mathbf{y}_n) + f(t_{n+1}, \mathbf{y}_n + hf(t_n, \mathbf{y}_n))) \tag{20}$$

So, there are two differences between the expansion rule with Heun's method:

- The expansion rule losses two identity mapping, namely $\mathbf{y}_n$, compared to Heun's method.
- FractalNet faces the same problem with ResNet that the step size $h = 1$. Thus, FractalNet cannot holds the upper bound given by Heun's method.

And step size $h = 1$ is the main reason that FractalNet fails to resist adversarial examples.

### 3.3.3 REVNET

RevNet (Gomez et al., 2017) introduced a variant of ResNet where each layer's activations can be reconstructed exactly from the next layer's. Each reversible block takes inputs and produces outputs according to the following additive coupling rules

$$
\begin{aligned}
X_{n+1} &= X_n + f_1\left(Y_n\right) \\
Y_{n+1} &= Y_n + f_2\left(X_{n+1}\right)
\end{aligned}
\tag{21}
$$

This additive coupling rules can be interpreted as Euler's method applies to the following systems of differential equations (Lu et al., 2018)

$$
\begin{aligned}
\dot{X} &= f_1(Y, t) \\
\dot{Y} &= f_2(X, t)
\end{aligned}
\tag{22}
$$

RevNet faces the same problem with ResNet that the step size $h = 1$. Thus, RevNet cannot holds the upper bound given by Euler's method for the dynamic systems.

## 4 EXPERIMENTAL RESULTS

### 4.1 RESNET WITH STEP SIZE $h \neq 1$

We first evaluate the performance of ResNets with different step sizes on adversarial examples. Adopting the same hyperparameter settings for ResNet56 that presented in He et al. (2016), we train 11 ResNet56s on CIFAR10 datasets with the step size decreases from $10^{-0}$ to $10^{-10}$.

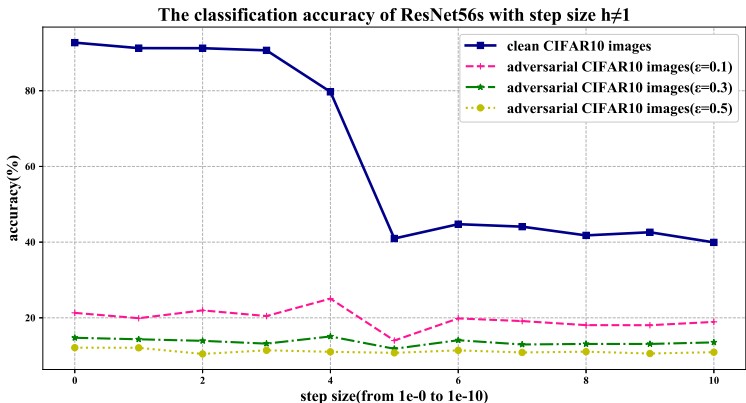

Figure 1: The classification accuracy of 11 ResNet56s both on clean CIFAR10 images and three kinds of adversarial CIFAR10 images(on the log scale). The adversarial CIFAR10 images are generated by Fast Gradient Sign Method with $\epsilon = 0.1, 0.3, 0.5$ on another neural network which has three residual blocks.

Through the numerical stability analysis of ResNet, we know that the classification result should be well behaved when considering small adversarial noises, provided that the step size of $h$ is sufficiently small. In actual computations, however, when we decrease the step size from $10^{-1}$ to $10^{-10}$, ResNet has no obvious robustness against adversarial examples. And when step size is very small, ResNet is difficult to train. Therefore, the accuracy of ResNet56 is considerably low both on clean and adversarial inputs. We consider that ResNet is impossible to have natural robustness against adversarial examples.

### 4.2 THE NATURAL ROBUSTNESS OF NEURAL ODES

As we mentioned in 3.2, Neural ODEs define a continuous dynamic system, which is equivalent to ResNet with step size $h \rightarrow 0$. And Neural ODEs hold a weaker upper bound to constrain the error

caused by adversarial noises. In this section, we evaluate the performance of Neural ODEs under three white-box attacks, FGSM(Goodfellow et al., 2014), PGD(Madry et al., 2017), DI$^2$-FGSM(Xie et al., 2019) and one black-box adversarial attack, Boundary Attack(Brendel et al., 2017).

Without data augmentation, the accuracy of ResNet on clean MNIST and CIFAR10 datasets is 99.33% and 89.60%, the accuracy of Neural ODEs on clean MNIST and CIFAR10 datasets is 99.05% and 69.94%. Experiment details can be seen in appendix A.

### 4.2.1 THE EVALUATION ON WHITE-BOX ADVERSARIAL ATTACKS

Table 1 shows that Neural ODEs have natural robustness under FGSM and PGD attacks. The classification accuracy of adversarial MNIST images generated by FGSM only drops by less than 2% for all $\epsilon$. Even the strongest first order attack method PGD can hardly beat Neural ODEs on MNIST. Although the performance of Neural ODEs on clean CIFAR10 images is worse than state-of-the-art networks like ResNet, it still works well on adversarial CIFAR10 images generated by PGD.

| Datasets | Attack | Models | $\epsilon = 0.2$ | $\epsilon = 0.3$ | $\epsilon = 0.4$ | $\epsilon = 0.5$ |
|---|---|---|---|---|---|---|
| MNIST | FGSM | ResNet50 | 25.20 | 10.95 | 9.80 | 9.75 |
| | | Neural ODEs | **97.50** | **97.46** | **97.52** | **97.48** |
| | PGD | ResNet50 | 0 | 0 | 0 | 0 |
| | | Neural ODEs | **93.62** | **90.55** | **84.64** | **75.23** |
| Datasets | Attack | Models | $\epsilon = 8/255$ | $\epsilon = 12/255$ | $\epsilon = 16/255$ | $\epsilon = 20/255$ |
| CIFAR | FGSM | ResNet50 | 4.65 | 4.16 | 3.60 | 3.26 |
| | | Neural ODEs | **60.70** | **60.65** | **60.67** | **60.67** |
| | PGD | ResNet50 | 0 | 0 | 0 | 0 |
| | | Neural ODEs | **59.15** | **56.93** | **53.58** | **49.53** |

Table 1: The classification accuracy of Neural ODEs and ResNet50 under FGSM and PGD attacks. For both MNIST and CIFAR10, we set the size of perturbation $\epsilon$ of PGD in an infinite norm sense, the size of PGD step is set to 0.01, the number of PGD steps is set to 40 and a uniform random perturbation is added before performing PGD.

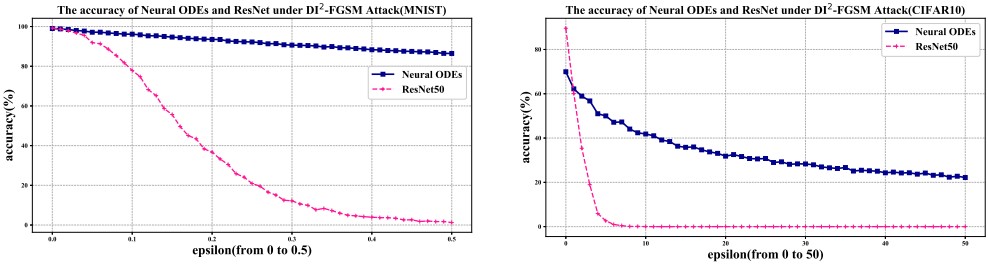

Figure 2: The evaluation of ResNet and Neural ODEs under DI$^2$-FGSM. $\epsilon = 0$ represents the accuracy of ResNet and Neural ODEs without the adversarial attack applied. For MNIST, the maximum perturbation $\epsilon$ is set to 0.5 among all experiments, with pixel value in [0,1]. For CIFAR10, the maximum perturbation $\epsilon$ is set to 50 among all experiments, with pixel value in [0,255].

As shown in Figure 2, when facing adversarial noises generated by DI$^2$-FGSM, for MNIST, Neural ODEs remains strongly resistant to perturbations while the accuracy of ResNet drops sharply with larger $\epsilon$. For CIFAR10, the reduction in accuracy of Neural ODEs is significantly less than ResNet50 with an increase of $\epsilon$. Overall, Neural ODEs is more stable when facing white-box adversarial noises compared with ResNet.

### 4.2.2 THE EVALUATION ON BLACK-BOX ADVERSARIAL ATTACKS

Boundary Attack is a decision-based attack that starts from a large adversarial perturbation and then seeks to reduce the perturbation while staying adversarial(Brendel et al., 2017). For MNIST, Boundary Attack has almost no effect on Neural ODEs. For CIFAR10, however, it seems that Neural

ODEs also fails in this gradient-free attack, but the performance of Neural ODEs is still better than ResNet50 when $\epsilon$ is larger than 33.

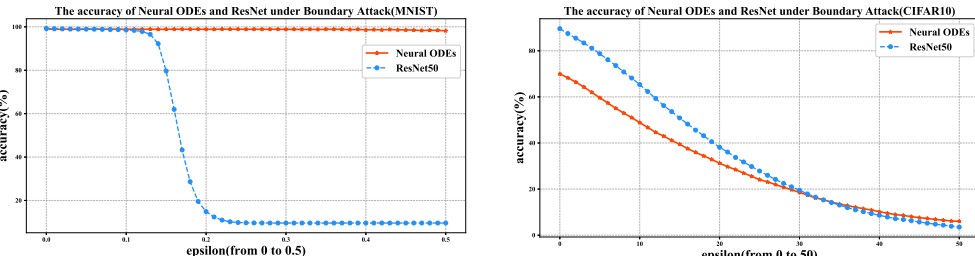

Figure 3: The evaluation of ResNet and Neural ODEs under Boundary Attack. For Boundary Attack, the number of queries is fixed at 1,000. The initial step size and orthogonal step size are set to 0.1.

### 4.2.3 COMPARING WITH ADVERSARIAL TRAINING METHODS

As one can see, Neural ODEs can remain its robustness and outperform PGD, TRADES, and YOPO without taking any adversarial training methods.

| Model | Adversarial training method | Clean | PGD-20($\epsilon = 8/255$) |
|---|---|---|---|
| PreAct-Res18 | PGD-10(Madry et al., 2017) | 84.82 | 41.61 |
| PreAct-Res18 | TRADES-10(Zhang et al., 2019b) | **86.14** | 44.50 |
| PreAct-Res18 | YOPO-5-3(Zhang et al., 2019a) | 83.99 | 44.72 |
| Neural ODEs | - | 69.94 | **59.06** |

Table 2: The comparison of robustness between Neural ODEs with PreAct-Res18 trained by three types of adversarial training methods on CIFAR10.

## 5 CONCLUSION AND DISCUSSION

In this paper, we highlighted and analyzed the natural robustness of Neural ODEs. We proved there are two similar upper bounds for ResNet and Neural ODEs under assumptions of continuity and step size $h \to 0$. We showed that it is the step size $h = 1$ causes ResNet and the other three deep neural networks with skip connections fail on adversarial examples. Neural ODEs define a continuous dynamic system with step size $h \to 0$. However, in actual computations, Modern ODE solvers adaptively adjust the step size to solve the continuous dynamic system and this brings us a weaker upper bound for Neural ODEs. We experimentally showed that Neural ODEs is more robust on adversarial examples compared with ResNet.

According to Theorem 3.1 and 3.2, the upper bound $c|\epsilon|$ and $\hat{c}|\epsilon|$ are related to Lipschitz constant and independent with the step size $h$. Cisse et al. (2017) showed that the robustness of DNNs can be improved by constraining the Lipschitz constant. In this paper, however, we are more concerned with the step size $h$, because the upper bound cannot be guaranteed if $h \to 0$ is not satisfied. We experimentally demonstrated that the difference of ResNet with Neural ODEs in step size $h$ is sufficient to make a significant difference in robustness, even without constraining the Lipschitz constant. Nevertheless, constraining the Lipschitz constant is expected to make neural networks to be more robust, which we would like to consider as our future work.

Due to the similarity between DNNs and numerical methods for solving ODEs, we believe that we can learn from the numerical stability analysis to design more robust deep neural networks.

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

# A    TRAINING CONFIGURATION OF NEURAL ODES

Table A1 summarizes our experiment settings for training Neural ODEs. We only change some hyperparameters in Neural ODEs, without taking any help of adversarial defense methods.

| Training Configurations | MNIST | CIFAR10 |
|---|---|---|
| training method | Adam | Adam |
| ODEslover | dopri5 | dopri5 |
| convolution layers | 4 | 5 |
| convolution channels | 128 | 256 |
| start time $t_0$ | 0 | 0 |
| stop time $t_1$ | 100 | 500 |
| training epochs | 100 | 200 |
| initial learning rate* | $10^{-3}$ | $10^{-4}$ |
| mini-batches size | 32 | 32 |

Table A1: The training configurations of Neural ODEs on MNIST and CIFAR10. *Every 50 epochs, the learning rate is reduced to half.

It is worth to mention that there are no common training configurations for Neural ODEs on CI-FAR10, Dupont et al. (2019) had reported that the accuracy of Neural ODEs on CIFAR10 test sets is 53.7%±0.2%, our experiment result is better than it.

