# OpenReview forum: "TOWARDS NATURAL ROBUSTNESS AGAINST ADVERSARIAL EXAMPLES"
_ICLR.cc/2021/Conference — Reject_

### Official Review · AnonReviewer3 · 2020-10-28
**Interesting viewpoint, but main claim not convincing due to deficiencies in comparison methodology**

**Rating:** 5
**Confidence:** 4

**Review:**

This paper offers an interesting viewpoint of adversarial robustness by comparing neural networks with skip connections such as ResNet with their Neural ODE counterparts. The authors analyze the different behaviors of the networks through their Lipschitz constants. They also try to support their claims that Neural ODEs are more robust due to their continuity (small step sizes) through experiments.

One issue I find with the claim that neural ODE is more robust against adversarial examples compared to other neural networks with skip connections is that it does not appear to be an apples-to-apples comparison. The authors try to demonstrate in Theorems 3.1 and 3.2 that Neural ODE has a smaller Lipschitz constant due to using a smaller step size, but it does not take into account the different representational capacity of the model when we use step size h=1 in ResNet. Unless the authors can show the neural ODE version and the finite step size ResNet computes the same/similar classes of functions, comparing the Lipschitz constant is not very helpful. The same comparison issue is also present in the experiments, where the authors use different architectures for neural ODE and ResNet (many fewer layers for neural ODEs). It is very difficult to draw conclusions on which neural network is more robust if they have different representational capacities.

Other than this main issue, there are also some limitations of the current work. It only analyzes neural networks with skip connections, whereas the phenomenon of adversarial attacks is general for many different neural networks. Also, judging from the accuracies under FSGM and PGD attacks, using a neural ODE cannot beat current state-of-art approaches of adversarial defense such as directing training with PGD examples.

As an aside, can the authors supply more details on how they generate adversarial examples for the neural ODE model?

Based on the above observations, I believe the current paper is not ready for publication in ICLR yet. The author should focus on improving on their core argument on why the factor of step size directly contributes to robustness of neural ODEs.

---

> ### Author Response · Authors · 2020-11-18
> **Thanks for your comments. Reply.**
>
> We are enthused that the reviewer finds the paper offers an interesting viewpoint of adversarial robustness.
>
> We address some of your comments below.
>
> 1."Unless the authors can show the neural ODE version and the finite step size ResNet computes the same/similar classes of functions, comparing the Lipschitz constant is not very helpful. The same comparison issue is also present in the experiments, where the authors use different architectures for neural ODE and ResNet (many fewer layers for neural ODEs). "
>
> We think that ResNets and Neural ODEs try to solve the same problem in different ways. ResNets specify a discrete sequence of hidden layers, while Neural ODEs parameterize the derivative of the hidden state using a neural network, i.e. Neural ODEs continuously transforms the state.
>
> In our paper, we use 4 (for MNIST) and 5 (for CIFAR10) convolutional layers to parameterize the derivative of the hidden state, but this does not mean the layers of Neural ODEs are 4 or 5. The definition of the depth of an ODE network is not clear and a related quantity is the number of evaluations of the hidden state dynamics required. (Chen et al. Neural Ordinary Differential Equations)
>
> Nonetheless, we thank you for pointing out the architecture of Neural ODEs. It would be interesting to try different ways to parameterize the derivative of the hidden state and see its performance on adversarial examples, which we would like to consider as future work.
>
> 2.“It only analyzes neural networks with skip connections, whereas the phenomenon of adversarial attacks is general for many different neural networks.”
>
> Although our paper only analyzes neural networks with skip connections, we believe that our paper still provides contributions that are different from existing works. First, we introduced and formalized the numerical stability analysis for DNNs with skip connections. Second, we found that the step size $h$ affects the stability of DNNs and it is the main reason that makes DNNs vulnerable to adversarial examples.
>
> 3.The details on generating adversarial examples for the neural ODEs
> For PGD, the magnitude of perturbation $\epsilon$ is set to [0.1, 0.2, 0.3, 0.4, 0.5]. The size of PGD step is 0.01, the number of PGD steps for one attack is 40, and the random start (whether or not to add a random perturbation before performing PGD) is TRUE.
>
> We will add the above parameter settings of PGD to the experiment section of our paper.

---

### Official Review · AnonReviewer4 · 2020-10-30
**ODE integrator error bounds intrinsically confer adversarial robustness to Neural ODEs**

**Rating:** 5
**Confidence:** 4

**Review:**

This paper uses a high-order ODE solver to take an $h=1$ step of a neural
network layer.  The mechanics of training with an ODE solver that uses
parameters to determine the dynamics of producing a layer output is previously
known.  This paper notes that outputs produced using an ODE integrator, wrt
adversarial inputs, have established error bounds. They demonstrate the
additional adversarial robustness of operating in a regime better respecting
such bounds.

I feel the paper is well written and clear.  The central theoretical idea is not a huge leap, but is
novel in the context of robust machine learning, imho. While presenting a simple, basic idea is
always nice, the paper left me unclear about whether the demonstration was primarily intended
to demonstrate a not terribly surprising theoretical prediction, or whether the technique would
be useful in practice.

They show many common networks with skip connections (resnet, etc.) correspond
to a forward ODE integration scheme with large step size $h=1$, whereas the
theoretical ODE adversarial bounds only hold as $h\rightarrow 0$.  They first show
that using fixed h corresponding to some fixed learning rate either does not
satisfy $h\sim 0$ or has too slow convergence, with no gain in adversarial
robustness.  So instead, they use a variable step size integrator instead, to
project forward to a larger $h$.

The idea and theory are simple and fairly well presented, and the demonstration
on a simple dataset nicely shows the benefit of this approach compared to the
original neural network.  However I felt a large number of significant things
were left out regarding choice of integrator.  Most obviously, Table 1 lacks
dopri parameters.  How does adversarial robustness depend on such parameter[s]?
What is the effect on execution time?

They only use one variable step size integrator.  They use a 4th and 5th order
variable step size integrator, to demonstrate the predicted natural ODE-based
robustness.  Several times I felt the presentation hinting that the low-order
of integration schemes is to blame; however, for layers with discontinuities
(relu), I naively expect lower order (and more evaluations?) might work out
just as well.

While the experiments constitute a simple demonstration, it still remained
difficult to judge the practical importance without some data comparing
execution time.  And once one begins to consider efficiency, a question
of what styles of integrator work well in practice would be nice (ex. gear
vs. Richardson vs. their one chosen integrator).

Even if the ODE method is expensive, compute time can be compared with another
ways to promote robustness, such as adversarial training.  A comment about
feasibility of using ODE method in conjunction with other robustness methods
could be made.

---

I have read the authors' comments.  The addition of the boundary attack experiment
was an excellent step; however, it underscores a requirement for further analysis to
understand *why*, apparently, in some cases the additional theoretical bound *fails* to
confer significant robustness.  The suggested "natural robustness" is only sometimes
present. Often clean accuracy is much reduced, so the method is not yet one I would consider
useful yet.  For me, understanding when the method works well (or not so well) would
bring this work out of the realm of interesting theoretical bounds into one of more
general interest.

---

> ### Author Response · Authors · 2020-11-25
> **Thank you for reviewing this paper and giving your useful comments.**
>
> We are enthused that the reviewer finds the paper is well written and clear, the idea and theory are simple and fairly well presented, and the demonstration on a simple dataset nicely shows the benefit of this approach compared to the original neural network.
>
> We address some of your comments below.
>
> 1. "the paper left me unclear about whether the demonstration was primarily intended to demonstrate a not terribly surprising theoretical prediction, or whether the technique would be useful in practice."
>
> In Theorem 3.1 and Theorem 3.2, we try to show that a small adversarial perturbation initial value of the problem leads to a small change in the solution, provided that the function $f(t,\mathbf{y};\mathbf{\theta})$ is continuous and the step size $h$ is sufficiently small. We experimentally verified that the difference of ResNet with Neural ODEs in step size $h$ is sufficient to make a significant difference in robustness. We think this is because we didn't find a proper way to deal with the step size of common networks with skip connections.
>
> 2. "Most obviously, Table 1 lacks dopri parameters. How does adversarial robustness depend on such parameter[s]? What is the effect on execution time?"
>
> The dopri5 parameters we used are as follows : atol=$10^{-3}$, rtol=$10^{-3}$, safety=0.9, ifactor=0.9, dfactor=0.2, max_num_steps=2$^{31}$-1.
>
> Thank you for bringing these two questions. We are also interested in how these parameters affect adversarial robustness, which we would like to consider as our future work. For now, we found that the execution time can affect adversarial robustness to some extent. The larger the execution time chosen, the more robust Neural ODEs will be. We are conducting an explanation for this phenomenon.
>
> 3."They only use one variable step size integrator. They use a 4th and 5th order variable step size integrator, to demonstrate the predicted natural ODE-based robustness. Several times I felt the presentation hinting that the low-order of integration schemes is to blame."
>
> We also tried to set ODE solvers as Euler and RK4, but the classification accuracy is much lower than dopri5 on clean CIFAR10 test sets.
>
> 4. "Even if the ODE method is expensive, compute time can be compared with another ways to promote robustness, such as adversarial training. A comment about feasibility of using ODE method in conjunction with other robustness methods could be made."
>
> Upon your comments, we added a comparison between Neural ODEs and neural networks that are trained by adversarial training methods. Neural ODEs can remain its robustness and outperform PGD, TRADES, and YOPO without taking any adversarial training methods. Revisions of the paper have been uploaded. Details can be seen in Table2. Unfortunately, we didn't find a proper way to compare the compute time of Neural ODEs with the training time of adversarial training.
>
> We think it is feasible to combine Neural ODEs with adversarial training methods, we would like to consider this as our future work.
>
> Thank you for all your constructive comments.

---

### Official Review · AnonReviewer2 · 2020-11-03
**The paper analyzes the robustness of ResNets and Neural ODEs, highlights limitations of ResNets and suggests to use Neural ODEs. Some important references are missing; technical novelty is minor.**

**Rating:** 3
**Confidence:** 5

**Review:**

**Update**: Thank you to the authors for addressing the comments and updating the paper. I decrease my rating from 4 to 3 as the original claims of the paper were disproved by the experiments with black-box attacks on CIFAR10 which showed that Neural ODEs offer little advantage over Resnets. I believe that more experiments are needed to demonstrate that Neural ODEs offer robustness advantages. For example, when increasing the computational budget for one step FGSM attack, the accuracy stays the same, which might indicate that Neural ODEs obfuscates the gradient. Experiments with gradient-free attacks also indicate that Neural ODEs obfuscate the gradient (PGD has higher robust accuracy compared to gradient-free attacks). The authors should do an extensive evaluation with stronger attacks (PGD with DLR loss or CW loss with multiple random restarts up to 100-1000, AutoPGD); the study of the gradient obfuscation (confirm that when $\epsilon=1.0$, the attacks can always succeed).

#### Summary
The paper analyzes the robustness of residual deep neural networks through its relationships to ordinary differential equations. It establishes bound on the output change. It tries to explain the lack of robustness of ResNets. It suggests that Neural ODEs addresses the limitations of residual mappings and are naturally robustness to adversarial perturbations. In the experiments, the authors compared Residual Network and Neural ODEs to confirm that Neural ODEs are naturally robust.

#### Concerns:
- Missing references, e.g. Towards Robust ResNet: A Small Step but a Giant Leap Zhang et al (shows the limitation of ResNet and searched for optimal $h$ using grid search).
- Limited technical novelty in the main theoretical results in Theorem 3.1. Similar bounds in terms of Lipschitz constant might be established elsewhere.
- One of the main claims of the paper is that ResNets are not robust because $h = 1$. However, $c$ depends both on $h$ and $K$. So, it is possible to improve the robustness of ResNets by reducing its Lipschitz constant. However, this was not discussed in the paper.
- The paper suggests using Neural ODEs, which was published elsewhere. Yet, the experimental evaluation is limited. Models should be evaluated against a wider range of attacks, e.g. black-box attacks and other attack norms ($l_{2}$-norm).
- The attack parameters are not specified, e.g. number of attack iterations, number of restarts.

#### Minor comments:
- Incorrect citation for adversarial training method: cited Ganin et al. instead of Goodfellow et. al.
- Unfortunately, both of these explanations seem to imply that adversarial examples are inevitable to be avoided by deep neural networks. - the structure of the sentence might be incorrect.
- In this paper, we attempt to utilize the natural property of neural networks to dense adversarial examples. - the sentence is hard to understand.

---

> ### Author Response · Authors · 2020-11-25
> **Thanks for your comments. Reply.**
>
> Thank you for your helpful comments on our paper. Revisions of the paper have been uploaded.
>
> We address some of your comments below.
>
> 1. "Missing references, e.g. Towards Robust ResNet: A Small Step but a Giant Leap Zhang et al "
>
> Zhang et al. proved that a small step factor $h$ can benefit its training and generalization robustness during backpropagation and forward propagation, respectively. However, we find that the small step size $h$ in most of their experiments is 0.1. In Section 4.3, they performed a grid search of $h$ from 0.001 to 20, but they didn't evaluate these neural networks with different step sizes on adversarial examples.
>
> In our paper, we proved that there is an upper bound for neural networks with identity mappings to constrain the error caused by adversarial noises. The upper bound $c|\epsilon|$ indicates that ResNets can be well behaved when considering small adversarial perturbations, provided that the step size $h$ is sufficiently small. So, we trained 11 ResNet56s on CIFAR10 datasets with the step size decreases from $10^{-0}$ to $10^{-10}$. We found that ResNet has no obvious robustness against adversarial examples when step size $h$ is small (such as $10^{−1}$, $10^{−2}$, $10^{−3}$), and ResNet is difficult to train when step size is very small  (such as $10^{−8}$, $10^{−9}$, $10^{−10}$). Thus, we think our experiments are completely different.
>
> 2. "Limited technical novelty in the main theoretical results in Theorem 3.1. Similar bounds in terms of Lipschitz constant might be established elsewhere. One of the main claims of the paper is that ResNets are not robust because $h=1$. However, $c$ depends both on $h$ and $K$. So, it is possible to improve the robustness of ResNets by reducing its Lipschitz constant. However, this was not discussed in the paper."
>
> Although the Lipschitz condition is stronger than necessary, it simplifies the proofs. According to Theorem 3.1 and 3.2, the upper bound $c|\epsilon|$ and $\widehat{c}|\epsilon|$ are related to Lipschitz constant and independent with the step size. In this paper, however, we are more concerned with the step size $h$, because the upper bound cannot be guaranteed if $h\rightarrow0$ is not satisfied. We experimentally demonstrated that the difference of ResNet with Neural ODEs in step size $h$ is sufficient to make a significant difference in robustness, even without constraining the Lipschitz constant. Constraining the Lipschitz constant is expected to make neural networks to be more robust, which we would like to consider as our future work.
>
> 3. "Yet, the experimental evaluation is limited. Models should be evaluated against a wider range of attacks, e.g. black-box attacks and other attack norms."
>
> We changed the epsilons of PGD attack to [8/255, 12/255, 16/255, 20/255] on CIFAR10. We found that previous epsilons are unreasonable for CIFAR10. The new classification accuracy on adversarial inputs generated by PGD can verify the natural robustness of Neural ODEs. Details can be seen in Table1.
>
> We added one more white-box attack (DI$^{2}$FGSM) and a black-box attack (Boundary Attack) to our experiments. We found that Neural ODEs has significant robustness on these white-box attacks. When facing adversarial inputs that generated by Boundary Attack, for MNIST, Boundary Attack has almost no effect on Neural ODEs, for CIFAR10, however, it seems that Neural ODEs also fails in this black-box attack, but the performance of Neural ODEs is still better than ResNet50 when $\epsilon$ is larger than 33.
>
> 4. "The attack parameters are not specified, e.g. number of attack iterations, number of restarts."
>
> In Table1, for both MNIST and CIFAR10, we set the size of perturbation $\epsilon$ of PGD in an infinite norm sense, the size of PGD step is set to 0.01, the number of PGD steps is set to 40, and a uniform random perturbation ($-\epsilon$ to $\epsilon$) is added before performing PGD. We added the above parameter settings of PGD to the experiment section of our paper.
>
> 5. Upon your comments, we fixed some minor issues.
>
> We changed the citation for adversarial training method to Goodfellow et. al.
>
> We changed this sentence (Unfortunately, both of these explanations seem to imply that adversarial examples are inevitable to be avoided by deep neural networks.) to "Unfortunately, both of these explanations seem to imply that adversarial examples are inevitable for deep neural networks."
>
> We changed this sentence (In this paper, we attempt to utilize the natural property of neural networks to dense adversarial examples.) to "In this paper, we attempt to utilize the natural property of neural networks to defense adversarial examples."
>
> We look forward to clarifications from you.

---

### Official Review · AnonReviewer5 · 2020-11-05
**Misleading title; needs much stronger evaluation**

**Rating:** 3
**Confidence:** 5

**Review:**

 Summary:

This paper uses theoretical grounding, starting with Lipshitz continuity-based assumptions on residual connections, to show why such architectures are more susceptible to adversarial inputs. In the process, the authors draw a parallel between these residual connections and neural ODEs, showing how the latter can circumvent the main reason that leads to adversarial susceptibility for the former. Finally, via empirical evaluations, they show how neural ODEs have "natural" robustness to adversarial examples: they have a non-trivial performance on adversarial inputs, despite not being explicitly trained for robustness.


##########################################################################

Reasons for score:

This draft in its current form lacks strong adversarial evaluation and makes strong claims without any experimental evidence. The title and the body of the paper suggest "natural adversarial robustness", but evaluated only against ($L_2$: this is my guess since it is not specified in the paper) PGD and FGSM (which, as has been seen recently in the literature, is not useful to see) attacks. Moreover, the Lipshitz-based assumptions in the proof for deriving conditions for residual connections seem a bit too strong. Keeping strong assumptions and weak empirical evaluations in mind, I feel this paper needs a lot more work before it can be considered for acceptance.

##########################################################################

Pros:

- The analysis of existing variants of network architectures that include residual connections seems interesting. It helps to see why these models are theoretically unfit to achieve adversarial robustness. Some empirical evaluation on these models would also have been nice to see: just to see how one is more/less problematic than the other.

##########################################################################

Cons:

- My biggest concern is the weak evaluation of the mentioned models. Even though the paper's title (and most places in the paper) talk about "natural robustness", evaluation is performed only against one kind of attack (FGSM is just PGD with the number of steps = 1) and that too for just one norm. Please include more extensive evaluation, perhaps like second-order gradient attacks, gradient-free attacks, and augmentation based adversarial attacks, or change the title to reflect your current findings.

- A large body of related works seems to have been missed out in this paper's literature review. For instance, there have been several works on Lipshitz-constraint-based robustness for neural networks are highly relevant, but seem to have been missed out ([example](https://arxiv.org/pdf/1704.08847.pdf), [another example](https://openreview.net/pdf?id=HkxAisC9FQ))

- The second paragraph sets up the flow of the paper to hint at "designing a deep neural network that has natural robustness", whereas the main focus of this paper goes to the extent of only evaluating existing architectures. Additionally, there is no clear evidence to suggest that adversarial robustness is even possible just via a well designed neural network.

- Residual connections are not limited to skipping only one connection (Eq 1), and using this as a base assumption should either be explicitly stated or worked out for the general case.

- The function $f$ is related to one specific layer and is thus parameterized by its associated weights $\theta_n$. However, Eq 7 talks about the Lipschitz continuity for the general layer. Does this mean that this model assumes that __all__ layers under consideration are Lipschitz continuous, that too with the same constant? In an ideal scenario where all layers somehow indeed, using the same constant $K$ would require taking the largest one of all layers, which can make it ridiculously large. I think this is a fatal flaw in the derivation process, and the authors should address it.

- In what norm are all these examples operating? What are the parameters for the PGD attack? Configuration details like the number of steps, step size, and the number of random restarts, can make a significant difference in evaluation metrics. Also, since these attacks include randomness (especially PGD), please run them multiple times and report mean/std values.

- Many numbers in Table 2 do not make any sense (my guess is this is because of the randomness in these attacks, which makes it even more important to have multiple runs). For instance, accuracy __increases** for neural ODE on CIFAR10 for FGSM when the perturbation budget is **increased**? Attack success rates should be strictly non-decreasing with increasing attack budgets since the adversary can copy-paste smaller-budget attacks and get at least the same attack success rate. The same problem holds for Resnet50 on CIFAR10 ($\epsilon=0.4$ vs $0.5$), neural ODE on MNIST $\epsilon=0.3$ to $0.4$.


Please address and clarify these cons.

##########################################################################

Minor issues:

- Page 1, last paragraph "...which has been used to solve ordinary differential equations". Reference missing.

- Section 2.1, last paragraph: "...rely on using external models". This statement is not true. Popular adversarial defense techniques like feature squeezing([ref](https://arxiv.org/pdf/1704.01155.pdf)) do not augment the dataset or use an external model.

- Ambiguity in notation: next to Eq 1, what does $N(h)$ signify?

- The move from Eq 7 to Eq 8 is a bit non-trivial: please add more steps in between to show the process explicitly

- Section 3.3.1, just above Eq 16: "It has been show that ...". Reference missing

- Just above Eq 18: "can be formally rewrite" -> "can be formally rewritten as". Also, the $(1+x)^{-1}$ Taylor series expansion holds only when $|x| <1$. Is that the case here, *i.e.* is $|hf|<1$ ?

- Section 3.3.2, just above Eq 20 "resembles the Runge-Kutta method of order 2". Reference missing

- Is the x-axis for Figure 1 on the log-scale? Please clarify Please do not refer to adversarial inputs as "adversarial test set", as it is likely to be confused with an adversarial test set that has been generated offline and is used for evaluation.

- What are the numbers in Table 2? Accuracy ($f(\hat{x}) = y$), or 1 - error success rate ($f(x) = f(\hat{x})$)? Please clarify.

---

> ### Author Response · Authors · 2020-11-25
> **Thanks for your comments. Reply.**
>
> Thank the reviewer for finding the analysis of existing variants of network architectures is interesting. We have conducted more experiments in order to answer your questions. We believe these extra experiments have improved the presentation of the paper. Revisions of the paper have been uploaded.
>
> We address some of your comments below.
>
> 1. "My biggest concern is the weak evaluation of the mentioned models...Please include more extensive evaluation, perhaps like second-order gradient attacks, gradient-free attacks, and augmentation based adversarial attacks."
>
> We changed the epsilons of PGD attack to [8/255, 12/255, 16/255, 20/255] on CIFAR10. We found that previous epsilons are unreasonable for CIFAR10. The new classification accuracy on adversarial inputs generated by PGD can verify the natural robustness of Neural ODEs. Details can be seen in Table1.
>
> Upon your comments, we added one more white-box attack (DI$^{2}$-FGSM) and a gradient-free attack (Boundary Attack). We found that Neural ODEs has significant robustness on these three white-box attacks. When facing adversarial inputs that generated by Boundary Attack, for MNIST, Boundary Attack has almost no effect on Neural ODEs, for CIFAR10, however, it seems that Neural ODEs also fails in this gradient-free attack, but the performance of Neural ODEs is still better than ResNet50 when $\epsilon$ is larger than 33.
>
> 2. The second paragraph sets up the flow of the paper to hint at "designing a deep neural network that has natural robustness"...
>
> We changed this sentence to "Meanwhile, it leads to a more important question: how to make deep neural networks have natural robustness so that they can get rid of malicious adversarial examples."
>
> There are similarities between DNNs and numerical methods for solving ODEs. Through the numerical stability analysis, we can figure out ResNet does not satisfy the assumption of continuity and the step size $h\rightarrow0$. We think that we didn't find a proper way to deal with the step size of DNNs with skip connections, while Neural ODEs satisfies the step size $h\rightarrow0$ so that Neural ODEs are more robust than ResNet. We also experimentally confirmed that the difference of ResNet with Neural ODEs in step size $h$ is sufficient to make a significant difference in robustness.
>
> 3. "Does this mean that this model assumes that all layers under consideration are Lipschitz continuous, that too with the same constant? In an ideal scenario where all layers somehow indeed, using the same constant $K$ would require taking the largest one of all layers, which can make it ridiculously large."
>
> Thank you for pointing out this problem. Although the Lipschitz condition is stronger than necessary, it simplifies the proofs. The constant $K$ indeed require taking the largest one, we fixed this issue in our paper. In this paper, however, we are more concerned with the step size $h$, because the upper bound cannot be guaranteed if $h\rightarrow0$ is not satisfied.
>
> 4. "In what norm are all these examples operating? What are the parameters for the PGD attack? "
>
> In Table1, for both MNIST and CIFAR10, we set the size of perturbation $\epsilon$ of PGD in an infinite norm sense, the size of PGD step is set to 0.01, the number of PGD steps is set to 40, and a uniform random perturbation ($-\epsilon$ to $\epsilon$) is added before performing PGD.  We added the above parameter settings of PGD to the experiment section of our paper.
>
> 5. "Ambiguity in notation: next to Eq 1, what does $N(h)$ signify?"
>
> N(h) means that $n$(the number of steps/layers) is changed as the step size $h$ changes. To make sure that $nh$ is always equal to $b - t_{0}$.
>
> 6. Upon your comments, we fixed some minor issues.
>
> We added three missing references.
>
> We fixed "...rely on using external models" to "In addition to augmenting datasets or modifying original neural networks, there exist adversarial defense methods that rely on using external models and detecting adversarial examples."
>
> We added "on the log scale" to the caption of Figure1 and fixed the refer to adversarial inputs as "adversarial images".
>
> The accuracy in our paper means classification accuracy $f(\hat{x})=y$.

---

### Official Review · AnonReviewer1 · 2020-11-06
**Review for Paper**

**Rating:** 3
**Confidence:** 4

**Review:**

The paper claims that neural ODEs are more robust to adversarial examples than ResNets and offers both empirical evidence and a theoretical explanation. However, I don't believe the theoretical analysis shows what is claimed; in fact, it doesn't seem to say anything about the magnitudes of Lipschitz constants or a relative comparison between them in the case of ResNets vs. neural ODEs. Below are the main issues with the analysis:

Claims:

In the statement of Thm. 3.1, it is claimed that in ResNets, the distance between z_n (output corresponding to perturbed input) and y_n (output corresponding to the original input) is bounded above by some constant times the amount of perturbation to the input. The statement does not upper bound the constant and only lower bounds it trivially with 0. Therefore, this says nothing about how robust or not robust the model is to adversarial examples -- if the constant is large, the model could be sensitive to adversarial examples; if it's small, it could be robust. So, this statement is unrelated to the main claims of the paper.

In the statement of Thm. 3.2, a very similar claim is made about neural ODEs, except that the constant in front of the magnitude of perturbation (\hat{c}) is different from the constant in Thm. 3.1 (c). No claims on the relative magnitudes of \hat{c} and c are made, and so nothing can be said about the relative robustness or lack thereof of neural ODEs compared to ResNets.

Proofs:

If I understand correctly, Thm. 3.1 is supposed to make a claim about ResNets, which is when n (the number of steps/layers) is held constant as the step size (h) changes. In this case, the last step of the proof (eqn. 11) is incorrect, since nh does not necessarily equal to b - t_0, i.e. when n is fixed and h decreases, Euler's method will not reach b.

On the other hand, Thm. 3.2 seems to aim at making a claim about neural ODEs, which is when n increases as h decreases. In this case, nh would be equal to b - t_0. However the claim that the distance between z_n and y_n would be upper bounded in the limit of h -> 0 is odd, because it holds even when no limit is taken: nh is always equal to b - t_0 regardless of how small or large h is. So this cannot serve as an explanation for the differences in behaviours between neural ODEs and ResNets (where h = 1).

Significance:

The theoretical analysis is trivial and is a simple application of the definition of Lipschitz continuity, which is assumed. Additionally it has nothing to do with the main claims of the paper.

Conclusion:

While the empirical phenomenon has been convincingly demonstrated and is intriguing, the theoretical analysis doesn't contain anything substantive and isn't much of an explanation. No other attempt at explaining this phenomenon (empirically or otherwise) was made other than the provided theoretical analysis. So I believe the paper is incomplete and will need substantially more effort at explaining the phenomenon.

---

> ### Author Response · Authors · 2020-11-17
> **Clarifications on Proofs and Lipschitz constants**
>
> We thank the reviewer for constructive feedback.
>
> Clarification on Proofs：
>
> By Thm. 3.1, we intend to prove that for ResNets, when taking smaller steps (which means decrease $h$)and add more layers (which means increase $n$), in the limit, the distance between $z_{n}$ and $y_{n}$ is bounded. So, in Thm. 3.1, $n$ (the number of steps/layers) is changed as the step size $h$ changes. Therefore $nh$ is always equal to $b - t_{0}$.
>
> We think that this may be our experiment (4.1 RESNET WITH STEP SIZE $h\neq1$) misleads you. In 4.1, we fixed $n$ and changed only $h$. This is because we are more interested in the impact of changing $h$ on network performance. (Since a very deep ResNet with a very small step size is hard to train.)
>
> Clarification on Lipschitz constants：
>
> We thank you for pointing out this problem. Although the Lipschitz condition is stronger than necessary, it simplifies the proofs. We agree that we did not discuss Lipschitz constants in this paper. It is really difficult to compute Lipschitz constants between two convolutional layers and it would be a very nice extension of the current analysis, which we would like to consider as future work.
>
> But in this paper, we are more concerned with the step size $h$. Although $c$ and $\widehat{c}$ is independent of $h$, we think the step size $h$ still plays an important role in the numerical stability analysis. The upper bound $c|\epsilon|$ indicates that ResNets can be well behaved when considering small adversarial perturbations, provided that the step size $h$ is sufficiently small. In actual computations, however, the step size $h$ cannot be too small since a very small step size decreases the efficiency of ResNets. Neural ODEs don't have this problem and this is the biggest difference between Neural ODEs and ResNets. Therefore, Neural ODEs are more stable when facing adversarial noises compared with ResNets.

---

### Author Response · Authors · 2020-11-25
**Revision Submitted. New experiments and discussions added.**

Dear reviewers:

We uploaded a revision to our paper according to your comments. The changes are:

1.We changed the epsilons of PGD attack to [8/255, 12/255, 16/255, 20/255] on CIFAR10. We found that previous epsilons are unreasonable for CIFAR10. The new classification accuracy on adversarial inputs generated by PGD can verify the natural robustness of Neural ODEs. Table1 shows the results. We added the parameter settings of PGD to the caption of Table1.

2.Addition of DI$^{2}$-FGSM attack (white-box) results in Figure 2. For MNIST, Neural ODEs remains strongly resistant to perturbations while the accuracy of ResNet drops sharply with larger $\epsilon$. For CIFAR10, the reduction in accuracy of Neural ODEs is significantly less than ResNet50 with an increase of $\epsilon$.

3.Addition of Boundary Attack (black-box) results in Figure 3. For MNIST, Boundary Attack has almost no effect on Neural ODEs. For CIFAR10, however, it seems that Neural ODEs also fails in this gradient-free attack, but the performance of Neural ODEs is still better than ResNet50 when $\epsilon$ is larger than 33.

4.Addition of comparison between Neural ODEs and adversarial training methods results in Table 2. Neural ODEs can remain its robustness and outperform PGD, TRADES, and YOPO without taking any adversarial training methods.

5.Discussion of Lipschitz constant in Section 5. According to Theorem 3.1 and 3.2, the upper bound $c|\epsilon|$ and $\widehat{c}|\epsilon|$ are related to Lipschitz constant and independent with the step size $h$. Cisse et al. showed that the robustness of DNNs can be improved by constraining the Lipschitz constant. In this paper, however, we are more concerned with the step size $h$, because the upper bound cannot be guaranteed if $h\rightarrow0$ is not satisfied. We experimentally demonstrated that the difference of ResNet with Neural ODEs in step size $h$ is sufficient to make a significant difference in robustness, even without constraining the Lipschitz constant. Constraining the Lipschitz constant is expected to make neural networks to be more robust, which we would like to consider as our future work.

6.Fix minor issues and typos.

Thank you all for your helpful comments.

---

### Decision · Program_Chairs · 2021-01-07
**Final Decision**

**Decision:**

Reject

**Comment:**

I thank the authors and reviewers for the discussions. Reviewers raised major concerns regarding the significance of the results and experiments. Given all, I think the paper needs more work before being accepted. I encourage authors to address comments raised by the reviewers to improve their paper.

- AC